# Sparse innervation and local heterogeneity in the vibrissal corticostriatal projection

**Kenza Amroune[1], Lorenzo Fontolan[2], Agnès Baude[1], David Robbe[2], Ingrid Bureau[1]***

[1]Aix-Marseille Université, INSERM, INMED, Marseille, France; [2]Aix-Marseille Université, INSERM, INMED, Turing Centre for Living Systems, Marseille, France

## eLife Assessment

This revised article provides **fundamental** findings on how the mouse barrel cortex connects to the dorsolateral striatum, uncovering that inputs from discrete whisker cortical columns are convergent and SPN-specific, but topographically organized at the population level. The evidence supporting this claim is **compelling**, demonstrating that SPNs uniquely integrate sparse input from variable stretches across the barrel cortex. The study would be of interest to basal ganglia and sensory-motor integration researchers.

***For correspondence:**
ingrid.bureau@inserm.fr

**Competing interest:** The authors declare that no competing interests exist.

**Abstract** The density and overlap of cortical axons in the dorsolateral striatum (DLS) have suggested that striatal neurons integrate widespread information from cortical regions that are functionally related. However, in vivo, DLS neuronal responses to sensory stimuli have shown unexpectedly high selectivity, raising questions about the actual degree of convergence of functional corticostriatal projections on individual striatal cells. Here, we investigated this question by focusing on the projections from different whisker cortical columns in mice, as they overlap in the striatum and are co-active during behavior. Using ex vivo patch-clamp recordings in the DLS and glutamate uncaging for focal stimulations in the barrel cortex, we were able to map the location of presynaptic neurons to individual striatal projection neurons (SPNs). We found that each SPN was innervated by cells located in a small number of whisker cortical columns scattered across the barrel field in the slice. Connectivity of single SPNs with cortical neurons was thus highly discontinuous horizontally, despite the presence of more potential connections. Moreover, connectivity patterns were specific to each cell, with neighboring SPNs sharing few common clusters of presynaptic cells in the cortex. Despite this sparse and distinct innervation of individual SPNs, the projection was topographically organized at the population level. Finally, we found similar innervation patterns for D1- and D2-type SPNs, but observed differences in synaptic strength in their connections with certain cortical layers, notably the associative layer 2/3. Our results suggest that the high convergence of somatosensory inputs to the striatum, enabled by diffuse and overlapping cortical innervation, is accomplished through sparse yet complementary connectivity to individual SPNs.

## Introduction

The dorsolateral striatum (DLS) receives dense glutamatergic innervation from all sensory cortical areas. Its presumed function is to monitor the animal's ongoing sensory information, contributing to the modulation of sensory-guided behaviors and the learning of voluntary tasks (*Pennartz et al., 2009*; *Znamenskiy and Zador, 2013*; *Sippy et al., 2015*; *Robbe, 2018*). The observations that

**Figure 1.** Potential connectivity patterns within the corticostriatal projection. A rich and overlapping cortical innervation of the striatum (lines) permits different connectivity patterns: Left, synapses are formed promiscuously. In this case, convergence is high as each striatal projection neuron (SPN) receives broad cortical input, from many cortical origins (colored bars on top). Middle and right, only a fraction of the potential connections are actually formed and, in these cases, convergence is lower. The selected presynaptic neurons are either scattered (middle) or topographically (right) positioned in the cortex. Increasing the selectivity of the corticostriatal connections while keeping their broad origin (middle) could generate maximal heterogeneity in the patterns of input for SPNs sharing the same striatal volume, due to a large number of combinations of cortical origins. In the case of a topographic innervation (right), neighboring SPNs form synapses with presynaptic neurons located in the same cortical region, reducing the local heterogeneity.

cortical neurons are overabundant compared to striatal projection neurons (SPNs) (*Oorschot, 1996*), that axonal projections originating from functionally linked cortical regions overlap in the striatum (*Flaherty and Graybiel, 1991*; *Alloway et al., 1999*; *Wright et al., 1999*; *Hooks et al., 2018*), and that synaptic glutamatergic inputs must overcome strong feedforward inhibition (*Mallet et al., 2005*; *Pidoux et al., 2011*) have led to the idea that striatal neurons integrate dense and broad cortical inputs. In fact, studies examining the density of axonal boutons relative to the single-cell dendritic spines have predicted that several thousand cortical neurons could potentially connect every SPN if synapses were formed promiscuously (*Kincaid et al., 1998*). However, if synapses were formed in a selective manner, this would reduce the degree of convergence of cortical input onto individual SPNs and potentially increase the heterogeneity of inputs between SPNs (*Kincaid et al., 1998*), a scenario with important implications for how cortical information is integrated in the striatum. These alternative connectivity models schematized in *Figure 1* remained to be tested functionally.

Here, we investigated this question at the level of the projections originating from the barrel field in the mouse primary somatosensory cortex (S1) and targeting the SPNs. Vibrissal information is transmitted to the cortical columns of S1 through parallel sensory channels, each corresponding to an individual whisker (reviewed in *Adibi, 2019*). In S1, layer 4 contains optically dense, barrel-shaped structures whose organization mirrors the spatial arrangement of whiskers on the snout. This feature provides a framework for studying neuronal connectivity in relation to the whisker cortical columns ex vivo. However, this model has not yet been used to examine connectivity within the functional vibrissal corticostriatal projections, leaving the degree of input convergence in this pathway unclear. On the one hand, axons of neurons located in different whisker cortical columns overlap heavily in the dorsal striatum, suggesting that SPNs could receive dense and broad vibrissal cortical input (*Wright et al., 1999*; *Alloway et al., 2000*). On the other hand, cell activity evoked by whisker deflection showed

notable heterogeneity in the DLS: neurons were either tuned to multiple whiskers, selective to a single whisker, or yet non-responsive (*Carelli and West, 1991*; *Pidoux et al., 2011*). These observations prompted us to investigate whether the vibrissal innervation of SPNs violated Peters' rule stating that connectivity is proportional to the overlap of axons and dendrites (*Peters, 1979*; *Braitenberg and Schüz, 1998*).

SPNs comprise two main populations, defined by their projection targets and expression of dopamine receptors: D1 receptors in the direct pathway and D2 receptors in the indirect pathway (*Smith et al., 1998*; *Kravitz et al., 2010*) (neurons are hereafter referred to as D1 and D2 SPNs). Electrophysiological studies have found that inputs received by D1 SPNs were stronger than those received by D2 SPNs, both in vivo and ex vivo (*Reig and Silberberg, 2014*; *Filipović et al., 2019*; *Kress et al., 2013*; *Parker et al., 2016*). These observations may indicate that inputs converge to different degrees to D1 and to D2 SPNs.

To investigate the connectivity and input convergence on SPNs, we used a functional mapping technique that combines patch-clamp recordings and laser scanning photostimulation (LSPS) with glutamate uncaging on a novel slice preparation, which preserved the organization of the vibrissal corticostriatal projections, from the superficial layers of the barrel cortex to the DLS. LSPS enabled precise mapping of corticostriatal functional connectivity by identifying cortical sites where stimulation evoked synaptic currents in the recorded SPNs, thereby localizing the cell bodies of their presynaptic neurons. This approach allowed us to determine both the cortical column and layer of origin within the barrel field in the slice for each SPN input. We found overall discrete innervations, as presynaptic neurons were thinly scattered across the barrel field, or present in a single cortical column. SPNs located nearby in the same slice had distinct patterns of innervation, suggesting that connections on each SPN were fewer than the local potential connections. No difference was found between the innervation patterns of D1 and D2 SPNs, but the strength of their connections with certain layers differed.

## Results

### A novel preparation for investigating functionally the organization of projections from the barrel cortex to individual SPNs in the DLS

To investigate the spatial organization of sensory projections to SPNs, we have developed a parasagittal somatosensory corticostriatal slice, which contained ~8 whisker barrels ('Materials and methods'; *Figure 2*, *Figure 2—figure supplement 1A*). Biocytin-labeled projections extended from the top of the barrel cortex to the DLS (*Figure 2A and B*). To further validate the angles used in the preparation, we virtually re-sectioned a brain from the Allen Mouse Connectivity Atlas in which pyramidal cells of the barrel cortex had been labeled in vivo (i.e., prior to slicing; *Figure 2—figure supplements 1B*). Cortical axons extended into the striatum in a similar manner in both the reconstituted and acute slices (compare *Figure 2A*, *Figure 2—figure supplement 1B*), supporting the idea that projections from the present cortical columns targeting the recorded striatal region were largely preserved in the acute slice. Animals were Drd1a-tdTomato hemizygous mice in which D1 receptor expressing neurons were labeled. D1$^+$ and D1$^-$ (D2$^+$) SPNs were recorded in the whole-cell voltage–clamp configuration (n=101 cells, N=54 mice). Simultaneously, an ultraviolet laser beam was directed at every site of a 29×16 pixel grid (2.1×1.1 mm, 75 µm spacing) to uncage glutamate over the barrel cortex, excite cortical neurons, and thereby reveal cells presynaptic to the recorded SPN (*Figure 2C and D*). When glutamate was uncaged on an excitatory cortical neuron innervating the recorded SPN, it elicited short-lived excitatory postsynaptic currents (EPSCs; <50 ms duration) (*Figure 2D*). We and others have shown that only subthreshold synaptic events are elicited with this method, implying that the evoked EPSCs are mono-synaptic and that feedforward inhibition is prevented (*Shepherd et al., 2003*; *Bureau et al., 2008*). In fact, glutamate uncaging excited cortical neurons above their firing threshold at a maximal distance of 75 µm from their cell body (*Figure 2E*, *Figure 2—figure supplement 2A*), on 2.8±0.3 sites of the LSPS grid (mean±sem; L5 pyramidal cells, n=40; 75 µm spacing grid; see 'Materials and methods'), consistent with a direct (i.e., non-synaptic) and nearly somatic excitation (data per cell type in 'Materials and methods' and *Figure 2*, *Figure 2—figure supplement 2*). Thus, the sites where LSPS evoked EPSCs in SPNs contained or were adjacent to the cell bodies of cortical neurons that were directly connected to the recorded cell. These sites are hereafter referred

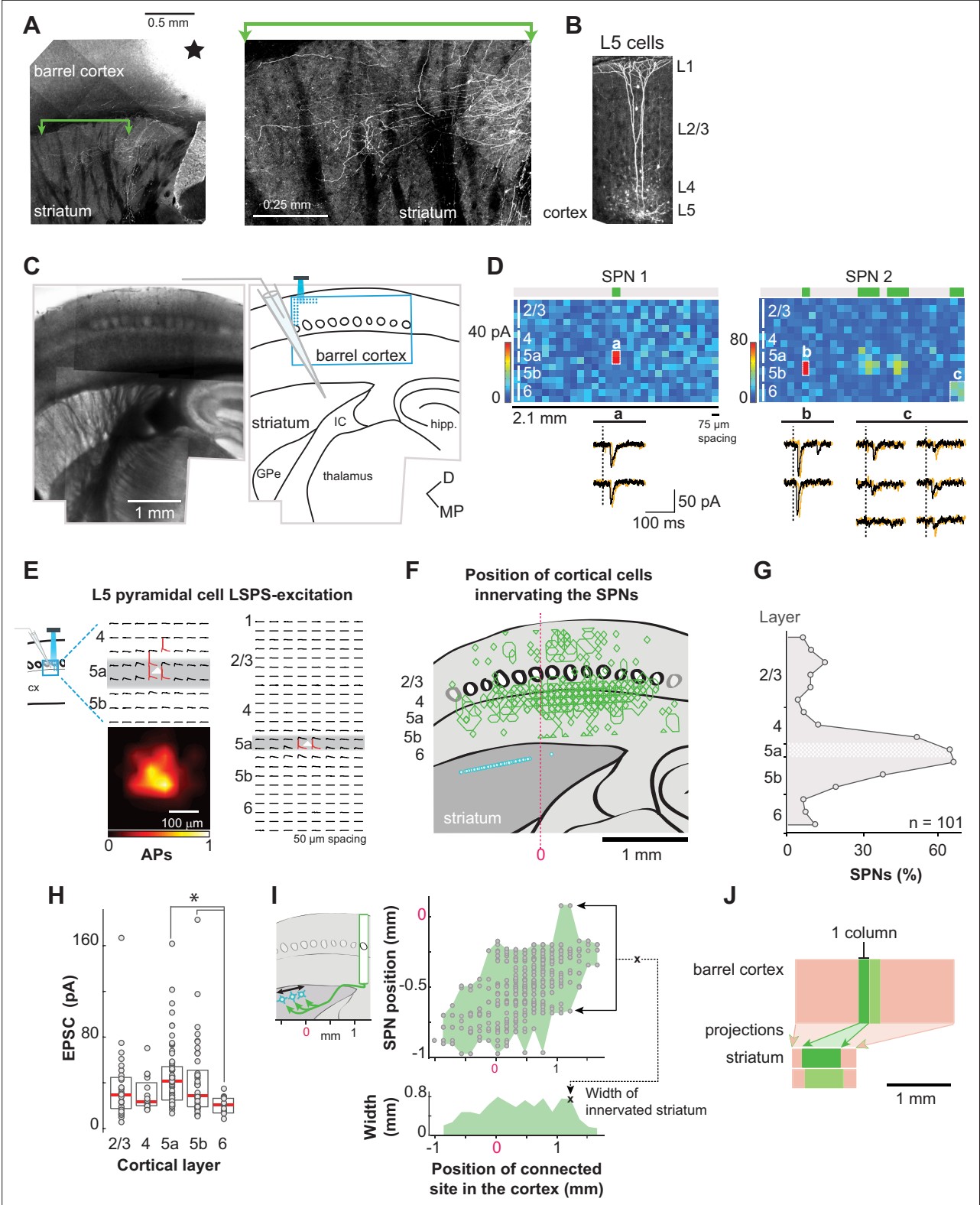

**Figure 2.** A novel slice preparation for investigating the spatial organization of the somatosensory projections to the dorsolateral striatum (DLS) neurons. (**A**) A somatosensory corticostriatal slice in which axons from L5 neurons were labeled with biocytin iontophoresis. Black star, the electrode position in L5a. Right, focus on a region in the striatum. (**B**) Right, Example dendritic arbors of L5 cortical cells labeled with biocytin in the slice. (**C**) Montage of a corticostriatal slice (left) and layout of the experiment (right). A striatal projection neuron (SPN) was recorded in the dorsolateral striatum while cortical neurons were photostimulated with laser scanning photostimulation (LSPS). The grid of LSPS (blue) was positioned on the barrel cortex.

*Figure 2 continued on next page*

*Figure 2 continued*

GPe, globus pallidus, external segment; IC, internal capsule; hipp., hippocampus. (**D**) Top, examples of synaptic input maps for individual SPNs showing one or four clusters of input (SPN 1 and 2, respectively). In the color maps, each pixel of color indicates the peak amplitude of excitatory postsynaptic currents (EPSCs) detected within a 50 ms window after the stimulus onset. The different cortical layers are represented by solid white vertical lines on the left side of the map. The green boxes at the top of the maps, the clusters of sites in the connectivity maps collapsed in the vertical axis whose stimulation evokes EPSCs. Bottom, EPSC traces evoked at the sites indicated by letters in the maps above. Two repetitions are superimposed (black and orange). Vertical dashed lines, the stimulus onsets (2 ms stimulus). (**E**) LSPS-evoked excitation of two L5a pyramidal neurons recorded in current-clamp mode. LSPS was an 8×8 (left) or 24×8 (right) 50 μm spacing grid. Cortical neurons are positioned at the center (white triangles). Traces with an action potential are in red. Bottom, average number of action potentials evoked at every site of the LSPS grid. (**F**) Overlay of the sites in the barrel cortex (green polygons) where stimulations evoked EPSCs in SPNs (cyan symbols; n=101 cells, N=54 mice). The red dashed vertical line is the reference (Ref$_{hor}$ in 'Materials and methods') used for aligning slices across experiments horizontally, on the junction of the striatum, GPe, and IC. (**G**) Contribution of each cortical layer to the SPN innervations. The 16 rows of the grid correspond to different cortical layers (layer 2/3: 1–6; L4: 7–9; L5a: 10; L5b: 11–13; L6: 14–16). The horizontal white band is L5a. (**H**) Amplitude of SPN EPSCs as a function of their laminar origin. Median (red) and 25–75th percentiles (boxes). * Kruskal–Wallis p=0.000125, Dunn–Šidák post hoc tests, p=0.0001 and 0.03415. (**I**) Top, positions of SPNs in the striatum for each position of connected sites on the horizontal axis of the LSPS grid. Bottom, maximal distance between SPNs for every connected site position in the barrel cortex on the horizontal axis, binned every 150 μm, in other words, the width of the projection zone of one cortical column within the striatum. (**J**) Schematic of the slice with the size of the cortical and striatal regions that are connected by projections to SPNs (pink) and the size of the striatal domain with functional projections from a single cortical column (shades of green).

The online version of this article includes the following figure supplement(s) for figure 2:

**Figure supplement 1.** Somatosensory corticostriatal slices generated from the Allen mouse brain atlas.

**Figure supplement 2.** LSPS-evoked excitation of the cortical pyramidal cells.

to as 'connected sites'. EPSCs had small amplitudes, 40±1 pA (n=550). Based on known amplitudes of spontaneous and miniature EPSCs in SPNs (10–20 pA on average; *Kreitzer and Malenka, 2007*; *Cepeda et al., 2008*; *Dehorter et al., 2011*; *Peixoto et al., 2016*), this finding is consistent with the presence of only one or a few presynaptic cells (≤5) at each connected site of the map.

For each SPN, an input map was assembled from the excitatory responses evoked at the uncaging sites of the LSPS grid (*Figure 2D*, top). From this input map, a binary map was derived that reported the presence or lack of evoked EPSC, referred to as 'connectivity map'. We first analyzed the ensemble of connectivity maps to characterize the global organization of the projections from the barrel field in the slice and checked that their properties were consistent with those described by anatomical studies (*Figure 2F*). When all SPN connectivity maps were superimposed and overlaid on the barrel cortex (see 'Materials and methods'; *Figure 2F*), the connected sites principally highlighted the layer (L) 5a. Indeed, ~65% of all SPNs received input from this layer (*Figure 2G*). Responses were elicited less frequently when stimulations were in L4, L5b, L2/3, and L6. We found that 50% of L5a cells fired action potentials when stimulations were at the bottom of L4, indicating that a fraction of the relatively high connectivity rate seen at the bottom of L4 was in fact with nearby L5a pyramidal cells (*Figure 2E*, *Figure 2—figure supplement 2A*). In contrast, photo-stimulations in L2/3 never elicited the firing of L5a cells (n=11; *Figure 2E*), indicating that all EPSCs evoked by stimulations in these superficial layers arose from direct synaptic connections between L2/3 pyramidal cells and SPNs. EPSCs evoked with stimulations in L2/3 to L5b had similar amplitudes (*Figure 2H*), suggesting that L5a dominance over these other layers is primarily due to a higher likelihood of SPNs being connected to it, rather than to stronger synaptic inputs. However, L6-EPSCs were smaller compared to L5a and L5b-EPSCs (Kruskal–Wallis, H(4) = 22.8, p=0.0001; Dunn post hoc test and Šidák correction, p=0.0001 and 0.034; *Figure 2H*). These observations are aligned with the anatomical studies that have identified L5a as the primary, although not exclusive, source of cortical innervation in the DLS (*Wise and Jones, 1977*; *Cowan and Wilson, 1994*; *Wall et al., 2013*).

In the superimposed connectivity maps, the ensemble of connected sites observed across recordings formed a band in the barrel cortex that was larger than the region in the DLS containing the recorded SPNs (2.5 mm vs. 1 mm, *Figure 2F*). An opposite asymmetry was observed when analyzing the striatal region connected to individual cortical columns: sites within a single cortical column (~150 μm in this preparation) were connected to SPNs distributed across a larger striatal region, spanning about 600 μm (*Figure 2I*). These two asymmetric relationships (converging vs. diverging), summarized in *Figure 2J*, implied that functional projections from adjacent cortical columns overlapped in the DLS, which is consistent with their known anatomy (*Wright et al., 1999*; *Alloway et al., 2000*; *Hooks et al., 2018*). Thus, projections from the cortical columns in the slice overlapped in the

striatum, which allowed us to investigate the degree of convergence of whisker cortical inputs onto individual SPNs.

## Low convergence of projections from the barrel cortex to individual SPNs in the DLS

To investigate the number and position of whisker cortical columns from which individual SPNs received synaptic inputs, each connectivity map was collapsed along its vertical axis and the horizontal distribution of connected sites was characterized using the following metrics (*Figure 3A*): (1) the input field, or the overall region within the barrel field in the slice from which an SPN receives inputs. Its width is the distance between the SPN most distant connected sites; (2) the cluster, a region in the collapsed connectivity map constituted of adjacent connected sites; (3) the cortical column, or its proxy: 1–2 adjacent connected sites in the collapsed connectivity map (i.e., 75–150 µm width). The connected sites were organized in 1–6 clusters, 1.9±0.1 on average in the input fields (*Figure 3B*). Typically, the width of clusters was about one cortical column (171±7 µm) and rarely exceeded two adjacent columns (*Figure 3C*). Finally, SPNs were innervated by cells distributed in close to 3 columns (2.7±0.2), though some SPNs with several clusters (54%) were innervated by as many as eight columns (*Figure 3D*). The distance between clusters was 75–900 µm, 265±28 µm on average (n=55; *Figure 3C*). Thus, SPNs could receive inputs from clusters of cortical cells separated by substantial connectivity gaps, often spanning several columns. Heterogeneous spacing and number of clusters contributed to the variability of the input fields width, which ranged between 0.075 and 1.6 mm (average, 535±42 µm; *Figure 3E*). Strikingly, regions lacking evoked synaptic responses (i.e., connectivity gaps) made up an average of 45% of the length of input fields with multiple clusters (maps collapsed along the vertical axis; *Figure 3F*, bottom). Moreover, the number of clusters was not increased proportionally in very large input fields, >1.4 mm, in which connectivity gaps amounted to more than 50% of space in the horizontal axis (*Figure 3F*, bottom). Altogether, these findings suggest a connectivity pattern characterized by a few small patches of presynaptic cortical cells scattered across the axis of barrels.

It could be that small input fields that contained a single cortical column (75–150 µm) were obtained when recordings were performed on the edges of the striatal region with cortical innervation in the slice. Contrary to this hypothesis, we found that SPNs with small input fields and those with broader ones were intermingled along the horizontal axis in the striatum (open and solid symbols in *Figure 3G*, respectively). In fact, the width of the input fields remained relatively constant across SPN positions in the slice (*Figure 3G*, top; Spearman correlation coefficient, *R*=0.05, p=0.61). This suggests that the position of our recordings did not impact the SPN input fields. Furthermore, SPNs receiving input from a single column were often located in slices where other cells received input from multiple ones (*Figure 3D*), reinforcing that the low functional connectivity with barrel columns in the slice was genuine in these cases.

Next, we investigated these connectivity patterns along their vertical axis. Individual SPNs had connected sites distributed in 2.4±0.1 cortical layers on average (2.9±0.1 for SPNs with several clusters). But these sites were not necessarily vertically aligned within a cortical column (examples in *Figures 2D and 3A*). In fact, in 55% of the cases, a column of the barrel field innervated an SPN through projections from a single cortical layer. This suggests that neurons projecting to an SPN are dispersed in the barrel cortex in such a way that the transmitted inputs may originate from distinct layers as well as from different whisker cortical columns. Our findings are summarized in *Figure 3H*, with schematics of the average and principal connectivity patterns.

## Individual SPNs have unique innervation

The above analysis has revealed great heterogeneity between SPN innervations. We then investigated whether innervations were more similar when SPNs were close to each other. To address this question, we compared the connectivity patterns of SPNs recorded in the same slice (*Figure 4A*). First, we observed that the majority of SPN pairs had their input fields overlapping in the barrel cortex (n=70; *Figure 4B*). The ratio was the highest, 77%, for SPNs that were neighbors in the striatum (<100 µm, n=33), but remained above 50% for cell distances of 400–500 µm. We next examined to what extent the connected sites were shared or vertically aligned between the connectivity maps. In contrast to input fields, alignment or overlap of connected sites was only observed when SPNs

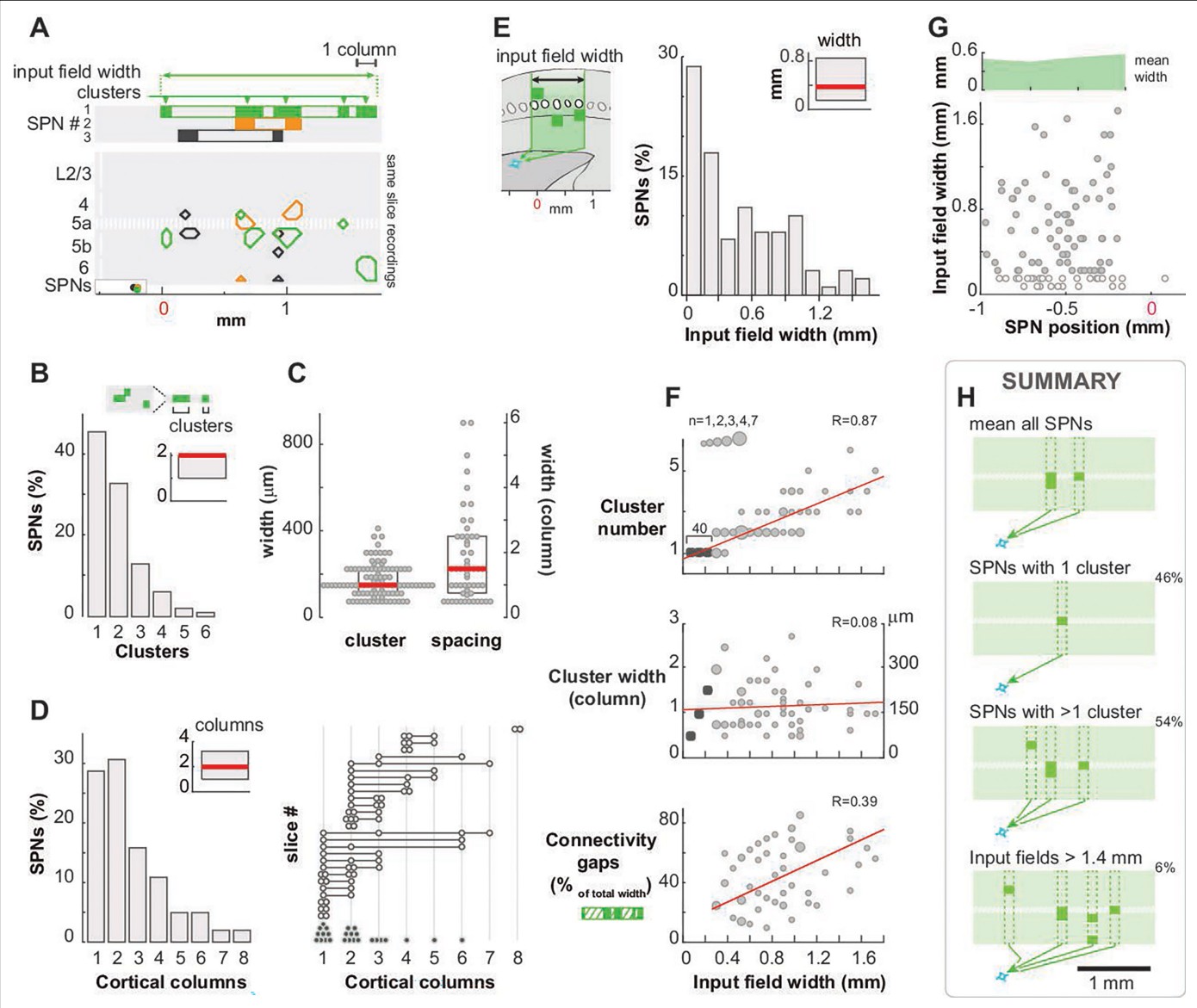

**Figure 3.** Sparse functional projections from the barrel cortex to individual striatal projection neurons (SPNs) in the dorsolateral striatum (DLS). (**A**) Example slice with three recordings. The hatched horizontal band is L5a, the polygons are the stimulation sites evoking excitatory postsynaptic currents (EPSCs), the solid boxes on top are the connectivity clusters, and the open rectangle the input field in the collapsed connectivity map. The circles at the bottom mark the positions of SPNs on the horizontal axis. Recordings are color-coded. (**B**) Fraction of SPNs receiving inputs from 1 to 6 clusters of projections in the barrel cortex. In the inset, median (red) and 25–75th percentiles (box). Clusters are defined as the ensemble of contiguous sites in the connectivity map collapsed in the vertical axis, whose stimulation evokes EPSCs (see examples in **A**, boxes at the top). n=101 cells, N=54 mice. (**C**) Median (red) and 25–75th percentiles (box) of cluster width and spacing, in μm (left Y axis) and in number of cortical columns (right Y axis). (**D**) Left, fraction of cells receiving inputs from 1 to 8 cortical columns in the barrel area. In the inset, median (red) and 25–75th percentiles (box). Right, number of cortical columns innervating individual SPNs. Symbols are cells, lines are slices. Light gray symbols, slices with ≥2 recordings. Dark symbols, one recording per slice. (**E**) Fraction of SPNs with input fields from 0.075 to 1.6 mm. (**F**) Top to bottom, for every input field width, the number of clusters, the cluster's width (in number of cortical columns and in μm), and the percentage of the region without connected site in the input fields collapsed along the vertical axis. Gray symbols, cells with ≥2 connectivity clusters. Other symbols, cells with one cluster. Top to bottom, p<0.0001, p=0.43, and p=0.0031 for the correlations. (**G**) Bottom, width of input fields in the barrel cortex for different positions of SPNs in the striatum. SPNs with an input field the size of a single or of several cortical columns are shown (open and solid symbols, respectively). Top, The average width of input fields as a function of SPN positions binned every 250 μm. (**H**) Schematized average and principal connectivity patterns. The % of SPNs is indicated for each pattern.

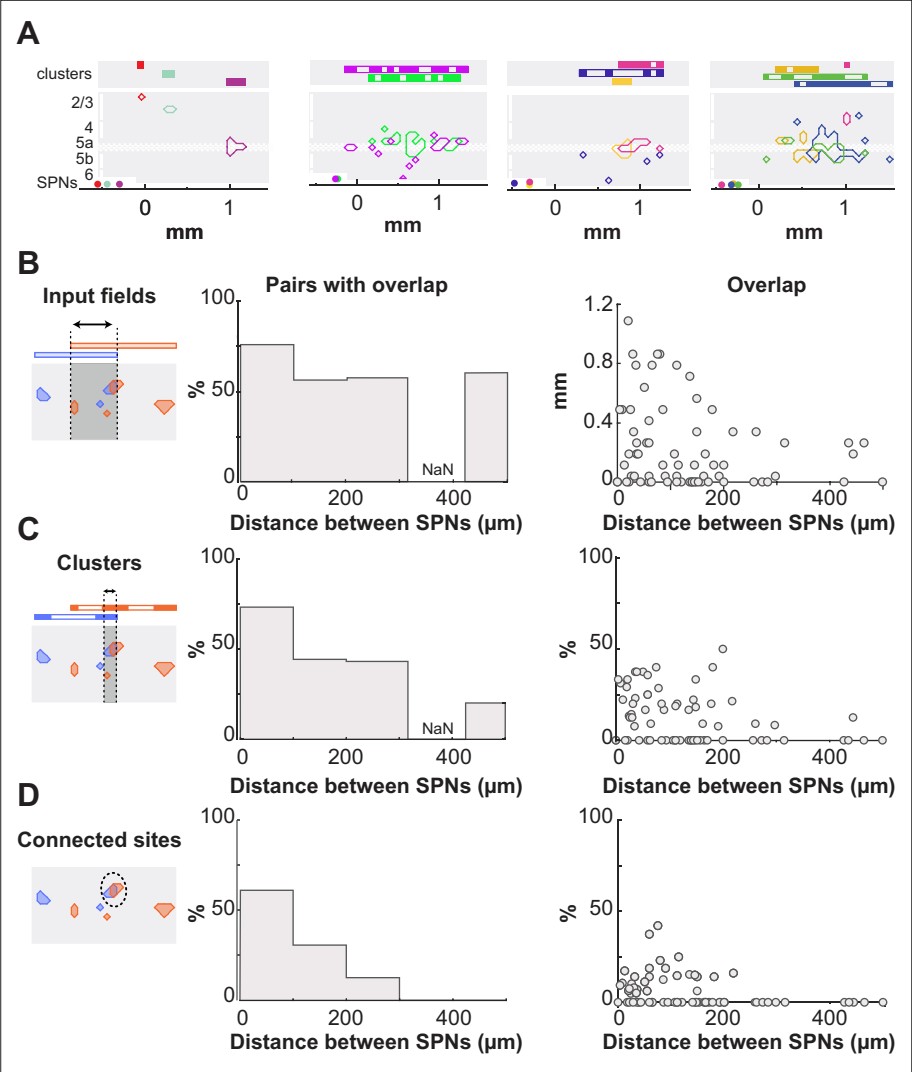

**Figure 4.** Individual striatal projection neurons (SPNs) have unique cortical innervation. (**A**) Four example slices with 2–4 recordings. Same as in *Figure 3A*. (**B**) Left, fraction of pairs with overlap in their input field. Right, the width of overlap as a function of the horizontal distance separating cells. n=70 pairs. NaN indicates the lack of data. (**C**) Same as in (**B**) for the fraction of vertical alignment between SPN connectivity maps. (**D**) Same as in (**B**) for the fraction of overlap between SPN connectivity maps.

were less than 300 µm apart (*Figure 4C and D*). Moreover, this concerned only a small fraction of the pair's connected sites: when SPNs were less than 100 µm apart, only 18±2% of the connected sites in the two connectivity maps were vertically aligned (*Figure 4C*) and 7.7± 1.6% of them overlapped (*Figure 4D*). Overall, these results indicate that DLS SPNs could receive inputs from the same domain in the barrel cortex and yet have patterns of cortical innervation without or little redundancy. They support a connectivity model in which synaptic connections on each SPNs are sparser than the potential connections, and with significant heterogeneity between SPNs in terms of their whisker-related cortical inputs.

## Topographic organization of SPN functional vibrissal innervation

The sparse innervation patterns, with limited similarity among neighboring SPNs, may seem inconsistent with the known topographic organization of somatosensory corticostriatal projections. To examine the global organization of functional projections in our preparation, we superimposed the connectivity maps of SPNs and labeled the connected sites based on the SPN's positions in the striatum (*Figure 5A*). SPNs located laterally/medially in the dorsal striatum had an input field whose

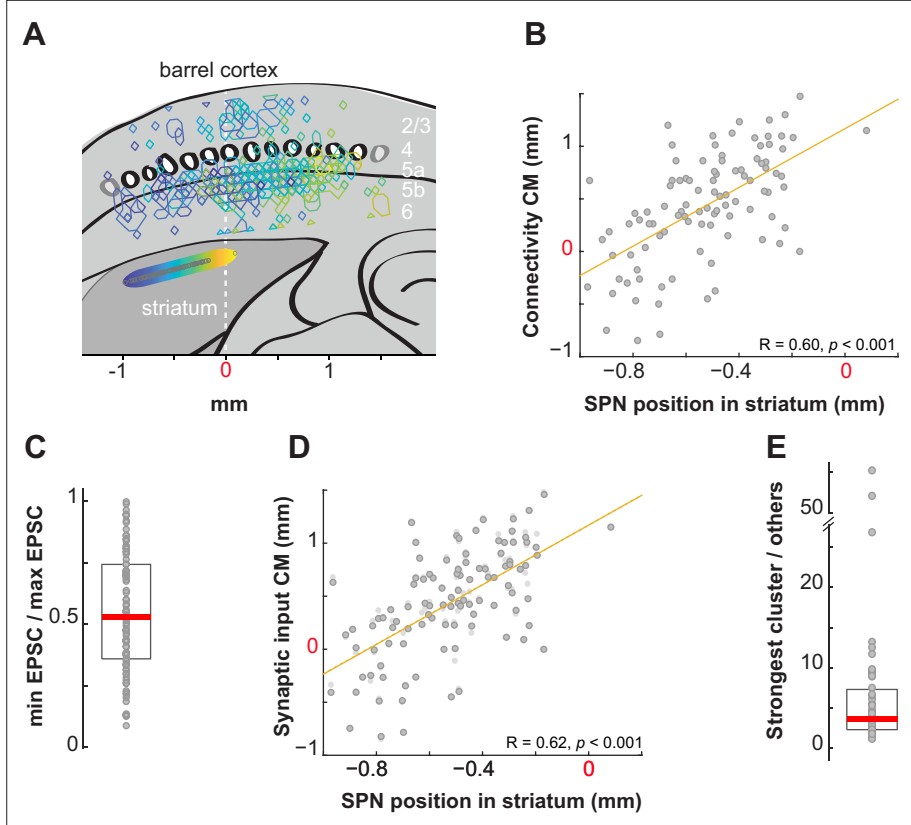

**Figure 5.** Topographic organization of the functional vibrissal innervation of striatal projection neurons (SPNs) in the dorsolateral striatum (DLS). (**A**) Overlay of the sites in the barrel cortex where stimulations evoked excitatory postsynaptic currents (EPSCs) in SPNs. The colors, yellow to blue, indicate the position of the SPNs (gray circles) along the horizontal axis in the striatum (axis at the bottom, 0 is Ref$_{hor}$ in 'Materials and methods'; n=101, N=54). Blue shades are for lateral SPNs. (**B**) Position of the connectivity center of mass (CM) as a function of the SPN position in dorsal striatum. Zero on the x and y axis is the position of the vertical dashed line shown in (A). (**C**) Ratio of the largest EPSC over the smallest EPSC for each recording. Median (red) and 25–75th percentiles (box). (**D**) Position of the synaptic input CM as a function of the SPN position in dorsal striatum. In light gray, the connectivity CM. (**E**) EPSC sum obtained in the strongest connectivity cluster relative to others' sum. Cells with ≥2 clusters, n=55.

connectivity center of mass (CM, see 'Materials and methods') was in the lateral/medial part of the barrel cortex (*Figure 5A and B*). Thus, the functional projections from the barrel cortex were topographically organized in the DLS, consistent with anatomy (*Flaherty and Graybiel, 1991*; *Alloway et al., 1999*; *Wright et al., 1999*; *Hooks et al., 2018*). Given the orientation of the slice (*Figure 2—figure supplements 1A*), a shift of the connectivity CM on the horizontal axis of the barrel field corresponded principally to a shift between whisker arcs (e.g., E1 to E8 barrel). The connectivity CMs of two adjacent SPNs could be 1 mm apart in the barrel cortex (*Figure 5B*), indicating that the topographic organization was not as precise as for intracortical projections (*Shepherd and Svoboda, 2005*; *Erlandson et al., 2015*). This finding, together with previous analyses, supports a connectivity model that falls between the two illustrated in the middle and right panels of *Figure 1* (selective, broad, and loosely topographic).

The EPSC amplitudes were heterogeneous within each input map (*Figure 5C*; ex *Figure 2D*). To test the hypothesis of a topographic organization influencing synaptic strength, we examined whether strong connections were associated with presynaptic cells located close to the connectivity CM. If such an organization of synaptic strengths existed, we would expect it to improve the correlation between the CM of synaptic input maps and SPN positions. However, we found nearly identical correlations whether taking the connectivity CM or the synaptic input CM, which is based on the connectivity pattern weighted by EPSC amplitudes (see 'Materials and methods'; R=0.60 vs 0.62; F(1) = 115.08,

p=0.98, ANCOVA; *Figure 5D*). While residuals from the correlation curve tended to be smaller using synaptic input CMs, the difference was not statistically significant (−9±7 μm, p=0.084, Wilcoxon, SPNs with >1 cluster). This was because synaptic input CMs were only slightly shifted from the connectivity CMs, by 45±5 μm for SPNs with two or more clusters (*Figure 5D*). These findings indicate that EPSC amplitudes showed no consistent pattern relative to the connectivity CM of the input field. This confirms the connectivity patterns as principal determinants of the topographic organization of somatosensory corticostriatal projections.

Finally, given that previous studies have revealed that responses of some striatal neurons exhibited strong selectivity for specific whiskers in vivo (*Carelli and West, 1991*; *Pidoux et al., 2011*), we investigated whether, in cases where SPNs were innervated by multiple clusters, one cluster might dominate the others. Comparing the sum of EPSCs corresponding to clusters, we found substantial variability, with the strongest cluster providing input that was 1.12–55 times greater than other clusters within the input field (×6.8 ± 1.4 on average; *Figure 5E*). Thus, connectivity patterns with multiple clusters may still permit some SPNs to have a preference for a particular whisker cortical column, due to the heterogeneity between clusters.

## D1 and D2 SPNs receive similar input from the barrel cortex

Finally, we investigated whether innervation patterns differed between the two populations of SPNs, expressing D1 or D2 receptors. Based on previous studies, unlabeled neurons in Drd1a-tdTomato hemizygous mice were principally SPNs expressing the D2 receptor (*Bertran-Gonzalez et al., 2008*; *Ade et al., 2011*; *Enoksson et al., 2012*; *Thibault et al., 2013*; *Cao et al., 2018*). As previously reported (*Wall et al., 2013*), projections to D1 and D2 SPNs were spatially intermingled along the axis of barrels (*Figure 6A and B*; n=47 and 54 cells, respectively). We found that D1 and D2 SPNs had similar input fields: first, their widths were not different (*Figure 6C*). In addition, they received input from a similar number of cortical columns, which were organized in clusters of similar widths (*Figure 6D and E*). Paired analysis of D1 and D2 SPNs recorded in the same slice also showed that their input fields did not significantly differ in width, number of cortical columns, or cluster width (p=0.44–0.50–0.26, Wilcoxon; *Figure 6C–E*). None of these parameters exhibited a clear relationship between subtypes ($R$=0.13–0.30–0.003; p=0.43–0.058–0.99, Pearson; *Figure 6C–E*). Finally, both types of SPNs received input from each cortical layer in similar proportions (*Figure 6F*). Thus, the heterogeneity in patterns of innervation is not due to a difference between SPN subtypes. Overall, there was no difference in the total inputs received by D1 and D2 SPNs (EPSC sum or EPSC$^T$; *Figure 6G*). In addition, the principal cluster was similarly dominant over the other clusters (*Figure 6H*). However, differences were observed in a layer-specific manner (*Figure 6I*). Whereas L5-EPSC$^T$ was similar in D1 and D2 cells, differences were found in L2/3 and L6: L2/3-EPSC$^T$ was larger in D2 SPNs (132±36 vs 53±22 pA, p=0.005, Mann–Whitney) but L6-EPSC$^T$ was larger in D1 SPNs (37±10 vs 16±2 pA, p=0.038). Further analysis suggested a difference in the strength of synapses as the principal mechanism. Indeed, L2/3-EPSCs evoked at single sites were larger in D2 SPNs (48±11 pA vs. 27±5 pA; p=0.043), whereas L6-EPSCs were larger in D1 SPNs (24±2 pA vs. 16±2 pA; p=0.038).

## Discussion

We investigated the organization of functional projections originating in the barrel cortex and targeting projection neurons in the dorsal striatum in a novel preparation. We found overall little input convergence as cortical neurons innervating an SPN were either located in a single cortical column or thinly scattered across the barrel field. Each SPN had a distinct pattern of innervation from the barrel cortex, albeit with occasional overlap for SPNs closely positioned in the DLS. D1 and D2 presented similar patterns of innervations.

### Sparse sensory innervation of single SPNs

As previously revealed by tracings of corticostriatal projections, the primary source of innervation of SPNs in this study is the upper part of L5 in the barrel cortex, L5a, although SPNs are occasionally innervated by cells located in deeper and superficial layers (*Wise and Jones, 1977*; *Reiner et al., 2003*; *Wall et al., 2013*; *Guo et al., 2017*; *Yamashita et al., 2018*; *Bertero et al., 2022*). Given the abundance of cortical axons in the striatum, it has been assumed that a large number of cortical cells

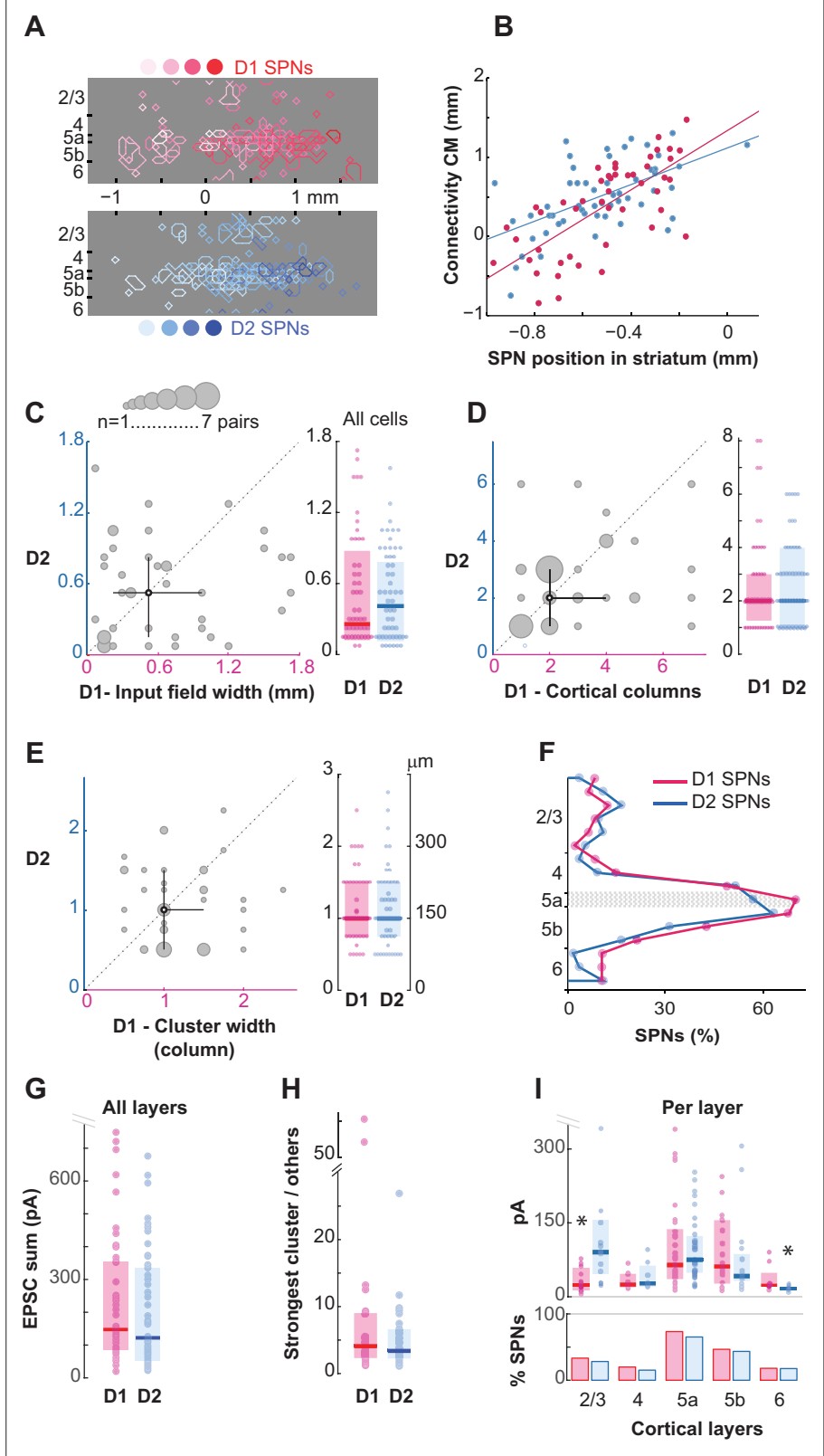

**Figure 6.** D1 and D2 striatal projection neurons (SPNs) have similar patterns of innervation from the barrel cortex.
(**A**) Overlay of the sites in the barrel cortex where stimulations evoked excitatory postsynaptic currents (EPSCs) in D1 (blue) or D2 SPNs (red). The shades, dark to light, indicate the position of the SPNs (bottom circles) along the medio-lateral axis in the dorsal striatum (axis at the bottom). D1 cells, n=47, N=36; D2 cells, n=54, N=37. (**B**)

*Figure 6 continued on next page*

*Figure 6 continued*

Position of the connectivity center of mass (CM) as a function of the SPN position in the dorsal striatum (D1, red; D2, blue). *R*=0.70 (D1, n=47, N=36); *R*=0.57 (D2, n=54, N=37). (**C**) Left, D2 SPN input field width as a function of the D1 SPN input field width, in the same slice. The black symbol and lines are the median value and 25–75th percentiles. Large symbols indicate n=2–3 pairs. n=42 pairs. Right, medians (thick lines) and 25–75th percentiles (boxes) of D1 and D2 input field widths, across all cells. n=101. (**D**) Same as in (**C**) for the number of cortical columns. (**E**) Same as in (**C**) for the width of connectivity clusters. (**F**) Contribution of each cortical layer to the SPN innervations. The horizontal hatched band is L5a. (**G**) EPSC sum for D1 and D2 SPNs. Outliers were not shown for clarity (1 D1, 2 D2, 1.2–1.5 nA). (**H**) EPSC sum obtained from the strongest cluster relative to others. Cells with ≥2 clusters. D1, n=24; D2, n=30. (**I**) Top, D1 and D2 SPN EPSC sum as a function of laminar origin. For each layer, only cells with inputs are included. Outliers are not shown for clarity (3 D1, 2 D2, 350–700 pA). * indicates a significant difference (p=0.005 and p=0.008; Mann-Whitney). Bottom, fraction of cells with input.

innervate each SPN, potentially up to a few thousand if synaptic connections were made promiscuously (*Kincaid et al., 1998*). On the other hand, it has also been reported that neurons in the DLS fire in response to the stimulation of a single body part in monkeys and rodents in vivo, even of a single whisker in some cases (*Carelli and West, 1991*; *Jaeger et al., 1995*; *Cho and West, 1997*). This finding suggested that each striatal neuron was in fact innervated by one small subregion of S1, although feedforward inhibition could have masked a broader selectivity (*Pidoux et al., 2011*). Here, the LSPS connectivity maps indicate that convergence from the barrel cortex to individual SPNs is indeed low as 60% of SPNs responded to the stimulation of only one or two whisker cortical columns present in the preparation. The LSPS combined with glutamate uncaging mapped projections contained in the slice, intact from the presynaptic cell bodies to the SPN dendrites. Some cortical inputs targeting distal SPN dendrites may have gone undetected either due to attenuation of synaptic events recorded at the soma or because distal dendritic branches were lost during slice preparation. Indeed, about 80% of S1 synaptic contacts are distributed along dendrites (*Sanabria et al., 2024*). However, synapses located distally are proportionally rare (*Sanabria et al., 2024*), and our estimates suggest that the loss of S1 input was minimal (see 'Materials and methods'). More significantly, our mapping only included projections from neuronal somata located within the S1 barrel field in the slice: projections from cortical columns outside the slice were not stimulated. For this reason, our study characterized connectivity patterns rather than the full extent of connectivity with the barrel cortex. A number of critical findings argue that innervation on individual SPNs is genuinely sparse: (i) the cortical columns present in the slice corresponded to different whisker arcs, an axis associated with the greatest overlap of cortical axons in the striatum (*Alloway et al., 1999*), and therefore where input convergence could have been high. This organization was visible in the global SPN LSPS connectivity

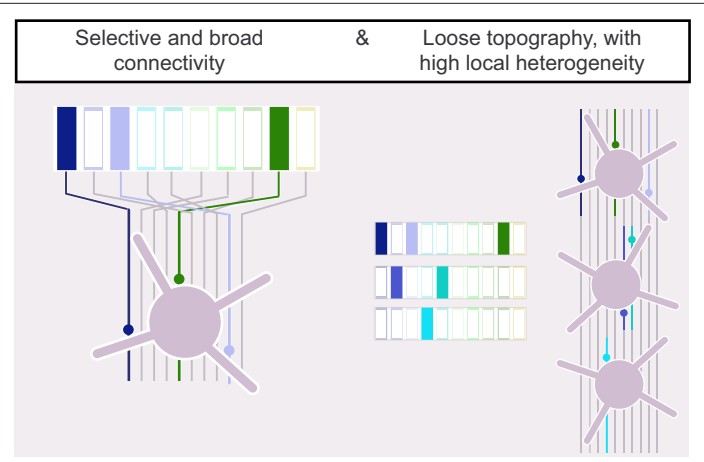

**Figure 7.** Connectivity pattern within the somatosensory corticostriatal projection to striatal projection neurons (SPNs). Left, our results support the model in which each SPN integrates limited and heterogeneous rather than exhaustive inputs transmitted by the barrel cortex, intermediate between the middle and right panels in *Figure 1*. Right, each SPN representation of the whisker cortical columns complements the representations of its neighbors.

map (*Figure 2J*). (ii) The peak of connectivity was with L5a as previously documented, not with a layer closer to striatum, as this would be the case if the native organization of the projection was altered in the preparation. (iii) The connected sites were scattered along broad horizontal and vertical axes, implying that projections in between were preserved. Critically, the fact that adjacent SPNs had different connectivity patterns in the same slice further supported the presence of a pool of potential connections larger than the number of actual connections made on each SPN. (iv) Connectivity clusters were sparser in larger input fields, suggesting that the number of presynaptic cells to an SPN was regulated and reached a plateau. (v) EPSCs were small, consistent with one or a few presynaptic cells at each connected site. Altogether, our results support the model in which, within a loosely topographical organization of projections, each SPN integrates limited and heterogeneous rather than exhaustive inputs transmitted by the barrel cortex (*Figure 7*). The speckled connectivity pattern of individual SPNs, arising from the abundant and diffuse cortical innervation in the DLS, suggests that somatosensory corticostriatal synapses are established through a selective and/or competitive process. It is important to determine whether this sparse innervation of SPNs by S1 is a characteristic shared with other projections. In particular, it will be interesting to test this hypothesis on the auditory projections targeting the posterior striatum, where neurons exhibit clear tone frequency selectivity (*Guo et al., 2018*).

## Specific cortical input to individual SPNs

A peculiar feature of the corticostriatal innervation is the imbalance between the number of cortical cells projecting to the striatum and the number of striatal cells, thought to be at a ratio of 10–1 (*Oorschot, 1996*). Based on this ratio and the ultrastructure of the corticostriatal connection, the density of axonal boutons and spines, C. Wilson's group concluded that the probability of an SPN being contacted by one given axon entering the domain of its dendritic arborization was low, 0.04–1.4%, depending on the model. Hence, the chance that two SPNs were innervated by the same cortical axon was even lower according to this model. Consistent with it, here, we found that overlaps between the connectivity maps of SPNs were rare and, when present, involved only a small fraction of the connected sites. This indicates that neighboring SPNs predominantly integrated distinct inputs from the barrel cortex, although it is possible that overlapping inputs received in distal dendrites were not all detected. Overlaps occurred only for SPNs located within 250 μm of each other. In contrast, input fields continued to intersect significantly beyond 250 μm and up to 500 μm of separation between SPNs. This means that relatively distant SPNs could share a general region of connectivity in the barrel cortex without any of their presynaptic partners residing within the same cortical column. Interestingly, the observation that two SPNs must be less than 250 μm apart to share some of their inputs was not predicted by the known ultrastructure of corticostriatal connections. In fact, it predicted the opposite: that two SPNs would be more likely to be innervated by the same cortical cell if their dendrites extended into different striatal subregions, as this would increase the likelihood of each SPN contacting boutons of this cortical cell, as these are sparsely distributed in the striatum (*Kincaid et al., 1998*). One possible explanation for this discrepancy could be that, in our experiments, pairs of SPNs with shared connectivity were innervated by different neurons that were co-localized in the barrel cortex (<75 μm apart). Such events could also be at the origin of the striatal clusters described in vivo, in which neurons had similar, yet non-identical, sensory selectivity (*Carelli and West, 1991*; *Jaeger et al., 1995*).

## Similarities and specificities in the patterns of projection to SPNs of the direct and indirect pathways

Electrophysiological studies have shown that D1 SPNs exhibit larger responses than D2 SPNs to whisker deflections (*Reig and Silberberg, 2014*; *Filipović et al., 2019*). Also, the optostimulation of cortical efferents in brain slices showed a bias toward D1 SPNs (*Kress et al., 2013*; *Parker et al., 2016*). In our study, we had contrasting results. There was no difference in the innervation of D1 and D2 SPNs, neither in the rate at which cells were connected by layers, nor in the organization of the projection. The principal layer innervating SPNs, L5a, delivered inputs of similar strengths to D1 and D2 SPNs. However, layers with lower incidence in the pattern of innervation showed bias in the amplitude of responses: stimulations in L2/3 induced stronger responses in D2 cells, whereas stimulations in L6 activated D1 cells more strongly. We have shown that stimulation in L2/3 activates the L2/3 projection only. Similarly, the L6-evoked EPSCs are consistent with the axon collaterals that L6 corticofugal

projection neurons have in the striatum (*Guo et al., 2017*; *Bertero et al., 2022*). The larger L2/3 inputs received by the D2 SPNs is intriguing because of the role of this layer in higher-order integration processes, such as those activated during operant sensory discrimination tasks (*Kwon et al., 2016*; *Han and Helmchen, 2024*; *Oryshchuk et al., 2024*). Moreover, the activity of L2/3 cells during mouse displacement is distinctive, exhibiting a sustained response to ongoing whisker-wall contacts, as opposed to the transient response observed in L5 (*Ayaz et al., 2019*). In L6, the population of corticofugal projection neurons is diverse, with some responding to whisking and sensory stimuli, while others are active during transitions into quiet periods of behavior (*Dash et al., 2022*; *Vélez-Fort et al., 2014*). Further investigation is needed to specifically examine projections from the upper and lower cortical layers and how they influence the dynamics of sensory integration in the direct and indirect pathways of the DLS. However, these data illustrate the diversity of paths of sensory integration involving D1 and D2 SPNs.

In conclusion, we found a low and sparse connectivity within the cortical vibrissal projections to individual SPNs. While projections respected a loose topography, the innervation patterns of individual SPNs displayed substantial connectivity gaps and heterogeneity. Since the inputs to a single SPN represent only a limited subset of whisker columns, a complete representation of whiskers could emerge at the population level, with each SPN's representation complementing those of its neighbors (*Figure 7*). These observations raise the hypothesis of a selective or competitive process underlying the formation of corticostriatal synapses. The degree of input convergence onto SPNs could be modulated by plasticity, potentially enabling experience-driven reconfiguration of S1 corticostriatal coupling.

## Materials and methods

**Key resources table**

| Reagent type (species) or resource | Designation | Source or reference | Identifiers | Additional information |
|---|---|---|---|---|
| Strain, strain background (*Mus musculus*) | B6.Cg-Tg(Drd1a-tdTomato)6Calak/J mice | Jackson Laboratories | 016204 | Hemizygous |
| Chemical compound, drug | MNI-caged-L-glutamate | Tocris | 1490/10 | 0.2 mM |

### Animals and ethics

Hemizygous male and female B6.Cg-Tg(Drd1a-tdTomato)6Calak/J mice (JAX stock #016204; *Ade et al., 2011*) were used on postnatal day 22–43. All of the animals were handled according to INSERM and French Ministry of Research guidelines. Protocols were approved by the committee #14 on the Ethics of Animal Experiments of the French Ministry of Research under the agreement APAFIS#27242.

### Brain slices preparation and electrophysiology

Mice were deeply anesthetized with isoflurane (4%) prior to cervical dislocation and decapitation. We prepared corticostriatal slices (350 μm thick) from the brain left hemisphere, based on stereotaxic coordinates placing the striatal cells 1–2 mm anteriorly to the projection neurons in the barrel cortex (*Aronoff et al., 2010*; *de la Torre-Martinez et al., 2023*). Parasagittal slices were cut with a 60° angle from the midline and a 10° angle in the dorso-ventral axis (*Figure 2—figure supplements 1A*) in a chilled cutting solution containing (in mM) 110 choline chloride, 25 NaHCO$_3$, 25 D-glucose, 11.6 sodium ascorbate, 7 MgCl$_2$, 3.1 sodium pyruvate, 2.5 KCl, 1.25 NaH$_2$PO$_4$, and 0.5 CaCl$_2$ (Sigma-Aldrich). Slices were then transferred to artificial cerebrospinal fluid (ACSF) containing (in mM) 127 NaCl, 25 NaHCO$_3$, 25 D-glucose, 2.5 KCl, 1 MgCl$_2$, 2 CaCl$_2$, and 1.25 NaH$_2$PO$_4$, aerated with 95% O$_2$/5% CO$_2$. Slices were first incubated at 34°C for 30 min and then maintained at room temperature for 20 min prior to use. Slices used for LSPS (1–2 per animal) contained barrels in the L4 of cortex, the globus pallidus, its external segment (GPe), the internal capsule, the ventral posteromedial nucleus of thalamus, and the anterior hippocampus. ~8 barrels (5–13) were visible in the slice. Two rows of barrels may have been superimposed. At the end of each experiment, a picture of the slice was saved in order to superimpose it digitally to other slices, according to visual landmarks (Photoshop; Adobe Inc). SPNs were visualized under infrared and fluorescent lights in a BX61WI microscope (Olympus) and patched with borosilicate electrodes (3–6 MΩ) and recorded in the voltage-clamp whole-cell configuration using a Multiclamp 700 A amplifier (Axon Instrument, Molecular Devices). The holding membrane

potential was –80 mV. The intracellular solution contained (in mM) 128 Cs-methylsulfate, 4 MgCl$_2$, 10 HEPES, 1 EGTA, 4 Na$_2$ATP, 0.4 Na$_2$GTP, 10 Na-phosphocreatine, 3 ascorbic acid; pH 7.25; 290–300 mOsm. Cells in L5 and L2/3 of the barrel cortex were recorded in the current-clamp mode, with an intracellular solution in which Cs-methylsulfate was replaced by K-methylsulfate. All experiments were performed at room temperature (21°C).

## LSPS with glutamate uncaging

LSPS was performed as described previously (*Bureau et al., 2006*). Recirculating (2 mL/min) ACSF solution contained (in mM) 0.2 MNI-caged-L-glutamate (Tocris), 0.005 CPP [()–3-(2-carboxypiperazin-4-yl) propyl-1-phosphonic acid], 4 CaCl$_2$, and 4 MgCl$_2$. Focal photolysis of caged glutamate was accomplished with a 2 ms 20 mW pulse of a pulsed UV (355 nm) laser (DPSS Lasers Inc) through a 0.16 NA 4× objective (Olympus). 25 mW laser pulses were used for stimulating cortical neurons in mice older than P30 to maintain their excitation at a similar level than in younger mice (*Figure 2*, *Figure 2—figure supplement 2B*). Ephus software for instrument control and acquisition (*Suter et al., 2010*) was used. The stimulus pattern for mapping the corticostriatal projections was 464 positions spaced by 75 µm on a 29×16 grid (2.1×1.1 mm) over barrel cortex. The corticostriatal slice and the LSPS grid were oriented in such a way that layer 5a was laid out horizontally. UV stimuli were applied every 700 ms and their successive positions on the LSPS grid were such as to maximize the time between stimulations of neighboring sites. Electrophysiological traces consisted of 100 ms baseline, a 450 ms window followed by a –5 mV 100 ms test pulse. A minimum of two and up to four stimulations were performed at each site at several minutes intervals. Excitation profiles of pyramidal neurons were generated under similar conditions except that cells were recorded in current-clamp mode and glutamate was uncaged on a smaller 8×8 grid covering their soma and dendrites (50 µm spacing; 350×350 µm). In a subset of L5a cell recordings, an 8×24 grid was used to stimulate up in L1 (50 µm spacing).

## Analysis of LSPS data

Synaptic input maps of neurons were constructed by taking the peak amplitude of EPSCs detected in a 50 ms time window starting at stimulation onset for each position in the LSPS grid. Measures were averaged across repetitions of stimulations (2–4). The threshold for EPSC detection was 3 standard deviations from baseline, or 9.2±0.4 pA. To disambiguate evoked responses from spontaneous activity, synaptic responses occurring less than two times across repetitions of maps were set to zero. Averaged maps were superimposed taking L5a as reference in the vertical axis and the junction of the GPe, dorsal striatum, and internal capsule as reference in the horizontal axis (Ref$_{hor}$). Pattern analyses were performed using custom software (*Bureau, 2025*) written in Matlab (MathWorks). In order to detect connectivity clusters and center of mass in the LSPS map, we used the binary version of the map ('connectivity map') reporting the location of connected and non-connected sites (i.e., yielding EPSCs or none in the recorded SPN). To detect clusters, the binary map was collapsed along its vertical axis. A cluster comprised one or more consecutive connected sites that was framed by one or more non-connected sites. Thus, a cluster here may include connected sites that were not adjacent on the vertical axis in the original map and may combine synaptic inputs from different layers. The number of cortical columns was estimated based on the number of consecutive connected sites (ConsSites) in the collapsed map and the mean width of a barrel in the slice, 150 µm, which is equal to two connected sites in the map. Hence, the number of cortical columns = $\Sigma$ (ceiling(ConsSites/2)). The connectivity center of mass was computed based on the uncollapsed map as follows: $\Sigma$ ($\Sigma_{vert}$ connected sites ×lateral distance from Ref$_{hor}$)/$\Sigma$ ($\Sigma_{vert}$ connected sites). The synaptic input center of mass was computed based on the original map, after thresholding, as follows: $\Sigma$ (mean$_{vert}$ EPSC ×lateral distance from Ref$_{hor}$)/$\Sigma$ (mean$_{vert}$ EPSC). Traces from current clamp recordings were analyzed to count the number of action potentials (APs) elicited upon glutamate uncaging. In the 8×8 grid, 50 µm spacing, total number of spikes was 2.2±0.2 for L2/3 pyramidal cells (n=26), 5.4±0.6 for regular spiking (RS) pyramidal cells in L5a (n=30) and 9.9±1.6 for non-RS pyramidal cells in L5b (n=10). The number of spikes elicited by uncaging at single sites was one or two and on average: 1.04±0.04 for L2/3 cells, 1.03±0.02 for L5 RS cells and 1.04±0.02 for L5 non-RS cells. As spacing in the stimulation grid was 50 µm in these particular experiments, we used the following equation to compute the number of sites with AP (sAP) in a stimulation grid with a spacing of 75 µm (i.e., as in connectivity maps): sAP$_{75}$ = sAP$_{50}$ × (75/50)$^2$. To estimate the loss of S1 synaptic contacts caused by

slice preparation, we modeled the SPN dendritic field as a sphere centered on the soma. S1 synapses were at 80% distributed radially along dendrites, according to the specific distribution described by *Sanabria et al., 2024*. The simulation also incorporated the known distribution of SPN dendritic length as a function of distance from the soma (*Gertler et al., 2008*). Finally, it was assumed that synapse placement was isotropic, with equal probability in all directions from the soma. Truncation was simulated by removing a spherical cap at one pole of the sphere, reflecting the depth of our recordings (beyond 80 µm). Based on this simulation, the loss of S1 inputs was <10%.

## Labeling of cortical cells

In a somatosensory corticostriatal slice prepared as described above, biocytin (2 %) was injected by iontophoresis in the L5a through a patch electrode (200 ms, 2.5 Hz) for 2×10 min (*Chang et al., 2000*). The slice was incubated for 4 h at room temperature in ACSF and transferred to 4% paraformaldehyde at 4°C overnight. It was rinsed in PBS containing 0.3% Triton X-100 (PBS-T), incubated overnight at room temperature in streptavidin-AlexaFluor488 (1:1000 in PBS-T) and mounted using Vectashield mount medium. Image acquisition was performed using a LSM 800 confocal Zeiss microscope.

## Statistical analysis

All data are expressed as mean ± SEM in the text. Median and 25–75th percentiles are shown in the figures. N and n are the numbers of animals and neurons, respectively. Unless stated otherwise, paired or unpaired non-parametric tests, and Kruskal–Wallis test for repeated measures followed by Dunn post hoc test and Šidák correction for multiple comparisons were used. $p < 0.05$ is considered statistically significant.

## Acknowledgements

This work was supported by funding from the Institut National de la Santé et de la Recherche Médicale and grants from the Agence Nationale de la Recherche (Corticostriatal, ANR-20-CE16-0002; STEP, ANR-20-CE17-0025-01). KA is supported by fellowships from the Ministère de l'Enseignement Supérieur et de la Recherche, from the governmental 'France 2030' program via A*Midex (Initiative d'Excellence d'Aix-Marseille Université, AMX-19-IET-004) and by ANR funding (ANR-17-EURE-0029). LF is supported by funding from the Excellence Initiative of Aix Marseille Université – A*MIDEX (Turing Centre for Living Systems). We thank the staff of the animal and genotyping facilities, the histology and imaging platforms of INMED, the members of the CBGB group and Antoine Depaulis for their help and support. We thank Elodie Fino, Rosa Cossart, Ede Rancz, Roustem Khazipov, Thomas Morvan, Jean-Luc Gaiarsa, Corinne Beurrier, and Lydia Kerkerian Le Goff for their critical reading of the manuscript.

## Additional information

### Funding

| Funder | Grant reference number | Author |
| --- | --- | --- |
| Agence Nationale de la Recherche | ANR-20-CE16-0002 | David Robbe Ingrid Bureau |
| Agence Nationale de la Recherche | ANR-17-EURE-0029 | Kenza Amroune |
| Agence Nationale de la Recherche | ANR-20-CE17-0025-01 | Ingrid Bureau |
| Aix-Marseille Université | A*MIDEX Turing Centre for Living Systems | Lorenzo Fontolan |
| Aix-Marseille Université | A*MIDEX AMX-19-IET-004 | Kenza Amroune |
| Ministère français de l'enseignement supérieur et de la recherche | | Kenza Amroune |

| Funder | Grant reference number | Author |
|---|---|---|
| Institut National de la Santé et de la Recherche Médicale | | David Robbe Ingrid Bureau |
| Centre National de la Recherche Scientifique | | Agnès Baude |

The funders had no role in study design, data collection and interpretation, or the decision to submit the work for publication.

### Author contributions
Kenza Amroune, Conceptualization, Formal analysis, Investigation, Visualization, Methodology, Writing – original draft, Writing – review and editing; Lorenzo Fontolan, Formal analysis, Writing – review and editing; Agnès Baude, Resources, Investigation, Writing – review and editing; David Robbe, Conceptualization, Supervision, Funding acquisition, Writing – review and editing; Ingrid Bureau, Conceptualization, Data curation, Software, Formal analysis, Supervision, Funding acquisition, Methodology, Writing – original draft, Project administration, Writing – review and editing

### Author ORCIDs
Kenza Amroune ⓘ https://orcid.org/0009-0004-4483-4741
Lorenzo Fontolan ⓘ https://orcid.org/0000-0002-1566-6636
Agnès Baude ⓘ https://orcid.org/0000-0002-7025-364X
David Robbe ⓘ https://orcid.org/0000-0002-9450-0553
Ingrid Bureau ⓘ https://orcid.org/0000-0001-9166-2250

### Ethics
All of the animals were handled according to INSERM and French Ministry of Research guidelines. Protocols were approved by the committee #14 on the Ethics of Animal Experiments of the French Ministry of Research under the agreement APAFIS#27242.

Reviewer #1 (Public review): https://doi.org/10.7554/eLife.106621.3.sa1
Reviewer #2 (Public review): https://doi.org/10.7554/eLife.106621.3.sa2
Reviewer #3 (Public review): https://doi.org/10.7554/eLife.106621.3.sa3
Author response https://doi.org/10.7554/eLife.106621.3.sa4

## Additional files

### Supplementary files
MDAR checklist

### Data availability
Electrophysiological data have been deposited at https://doi.org/10.57745/NPZKMQ and are publicly available. The code for the analyses presented in this paper is openly accessible at https://fr.math-works.com/matlabcentral/fileexchange/182036-connectivityanalysis (*Bureau, 2025*).

The following dataset was generated:

| Author(s) | Year | Dataset title | Dataset URL | Database and Identifier |
|---|---|---|---|---|
| Bureau I, Amroune K | 2025 | Données de réplication pour : Sparse innervation and local heterogeneity in the vibrissal corticostriatal projection | https://doi.org/10.57745/NPZKMQ | Recherche Data Gouv, 10.57745/NPZKMQ |

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
