## [Editor Report · eLife Assessment]

This revised article provides **fundamental** findings on how the mouse barrel cortex connects to the dorsolateral striatum, uncovering that inputs from discrete whisker cortical columns are convergent and SPN-specific, but topographically organized at the population level. The evidence supporting this claim is **compelling**, demonstrating that SPNs uniquely integrate sparse input from variable stretches across the barrel cortex. The study would be of interest to basal ganglia and sensory-motor integration researchers.

---

## [Referee Report · Reviewer #1 (Public review)]

Summary:

By applying the laser scanning photostimulation (LSPS) approach to a novel slice preparation, the authors aimed to study the degree of convergence and divergence of cortical inputs to individual striatal projection neurons (SPNs).

Strengths:

The experiments were well-designed and conducted, and data analysis was thorough. The manuscript was well written and related work in the literature was properly discussed. This work has the potential to advance our understanding of how sensory inputs are integrated into the striatal circuits.

---

## [Referee Report · Reviewer #2 (Public review)]

Summary:

How corticostriatal synaptic connectivity gives rise to SPN encoding of sensory information is an important and currently unanswered question. The authors utilize a clever slice preparation in combination with electrophysiology and glutamate uncaging to dissect the synaptic connectivity between barrel cortex and individual striatal SPNs. In addition to mapping connectivity across major anatomical axes and cortical layers, the authors provide data showing that SPNs uniquely integrate sparse input from variable stretches across barrel cortex.

Strengths:

The methodology shows impressive rigor and the data robustly support the authors conclusions. Overall, the manuscript addresses its core question, provides valuable insights into corticostriatal architecture, and is a welcomed addition to the field.

---

## [Referee Report · Reviewer #3 (Public review)]

Summary:

The authors explored how individual dorsolateral striatum (DLS) spiny projection neurons (SPNs) receive functional input from whisker-related cortical columns. The authors developed and validated a novel slice preparation and method to which they applied rigorous functional mapping and thorough analysis. They found that individual SPNs were driven by sparse, scattered cortical clusters. Interestingly, while the cortical input fields of nearby SPNs had some degree of overlap, connectivity per SPN was largely distinct. Despite sparse, heterogeneous connectivity, topographical organization was identified. The authors lastly compared direct (D1) vs. indirect (D2) pathway cells, concluding that overall connectivity patterns were the same, but D1 cells received stronger input from L6 and D2 cells from L2/3. The paper thoughtfully addresses the question of whether barrel cortex broadly or selectively innervates SPNs. Their results indicate selective input that is loosely topographic. Their work deepens the understanding of how whisker-related somatosensory signals can drive striatal neurons.

Strengths:

Overall this is a carefully conducted study, and the major claims are well-supported. The use of a novel ex vivo slice prep that keeps relevant corticostriatal projections intact allows for careful mapping of the barrel cortex to dorsolateral striatum SPNs. Careful reporting of both columnar and layer position, as well as postsynaptic SPN type (D1 or D2) allows the authors to uncover novel details about how the dorsolateral striatum represents whisker-related sensory information.

Weaknesses:

Most technical weaknesses have now been addressed in the text.

---

## [Author Response]

The following is the authors’ response to the original reviews.

**Reviewer #1 (Public review):**
This work focuses on the connection strength of the corticostriatal projections, without considering the involvement of synaptic plasticity in sensory integration.

Thank you for raising this point. Indeed, sensory integration is a complex process with a multitude of factors beyond connectivity patterns and synaptic strength. In addition, it is true that both connectivity levels and synaptic strength can be modified by plasticity.

We modified our conclusion as follows, line 354:

“Since the inputs to a single SPN represent only a limited subset of whisker columns, a complete representation of whiskers could emerge at the population level, with each SPN’s representation complementing those of its neighbors (Fig. 7). These observations raise the hypothesis of a selective or competitive process underlying the formation of corticostriatal synapses. The degree of input convergence onto SPNs could be modulated by plasticity, potentially enabling experience-driven reconfiguration of S1 corticostriatal coupling. “

**Reviewer #2 (Public review):**
A few minor changes to the figures and text could be made to improve clarity.

We thank you for having taken the time to indicate where changes could benefit the paper. We followed your recommendations.

**Reviewer #3 (Public review):**
(1) Several factors may contribute to an underestimation of barrel cortex inputs to SPNs (and thus an overestimate of the input heterogeneity among SPNs). First, by virtue of the experiments being performed in an acute slice prep, it is probable that portions of recorded SPN dendritic trees have been dissected (in an operationally consistent anatomical orientation). If afferents happen to systematically target the rostral/caudal projections of SPN dendritic fields, these inputs could be missed. Similarly, the dendritic locations of presynaptic cortical inputs remain unknown (e.g., do some inputs preferentially target distal vs proximal dendritic positions?). As synaptic connectivity was inferred from somatic recordings, it's likely that inputs targeting the proximal dendritic arbor are the ones most efficiently detected. Mapping the dendritic organization of synapses is beyond the scope of this work, but these points could be broached in the text.

Thank you for this analysis. The positions of S1 spines have been mapped on the SPN dendritic arbor by the group of Margolis (B.D. Sanabria et al., ENeuro 2024,10.1523/ENEURO.0503-23.2023). They observed that S1 spines were at 80 % on dendrites but with a specific distribution, on average rather close to the soma. In this study, S1 spines did not exhibit a specific distribution that would systematically hinder their detection in a slice. But, it remains that the position in the dendritic arbor where an S1 input is received does indeed impact its detection in somatic recordings. We modified the discussion as follows, line 275:

“The LSPS combined with glutamate uncaging mapped projections contained in the slice, intact from the presynaptic cell bodies to the SPN dendrites. Some cortical inputs targeting distal SPN dendrites may have gone undetected, either due to attenuation of synaptic events recorded at the soma or because distal dendritic branches were lost during slice preparation. Indeed, about 80 % of S1 synaptic contacts are distributed along dendrites (Sanabria et al., 2024). However, synapses located distally are proportionally rare (Sanabria et al., 2024), and our estimates suggest that the loss of S1 input was minimal (see Methods). More significantly, our mapping only included projections from neuronal somata located within the S1 barrel field in the slice: projections from cortical columns outside the slice were not stimulated. For this reason, our study characterized connectivity patterns rather than the full extent of connectivity with the barrel cortex.”

We explain our estimation of truncated S1 contacts in the Methods, line 434:

“To estimate the loss of S1 synaptic contacts caused by slice preparation, we modeled the SPN dendritic field as a sphere centered on the soma. S1 synapses were at 80 % distributed radially along dendrites, according to the specific distribution described by Sanabria et al. (2024). The simulation also incorporated the known distribution of SPN dendritic length as a function of distance from the soma (Gertler et al., 2008). Finally, it assumed that synapse placement was isotropic, with equal probability in all directions from the soma. Truncation was simulated by removing a spherical cap at one pole of the sphere, reflecting the depth of our recordings (beyond 80 μm). Based on this simulation, the loss of S1 inputs was < 10 %.”

(2) In general, how specific (or generalizable) is the observed SPN-specific convergence of cortical barrel cortex projections in the dorsolateral striatum? In other words, does a similar cortical stimulation protocol targeted to a non-barrel sensory (or motor) cortex region produce similar SPN-specific innervation patterns in the dorsolateral striatum?

This is an interesting question that could be addressed using the LSPS approach in areas for which ex vivo preparations have been designed to maintain the integrity of the corticostriatal projections, such as A1, M1 and S2.

We included this point in the discussion, line 299:

” The speckled connectivity pattern of individual SPNs, arising from the abundant and diffuse cortical innervation in the DLS, suggests that somatosensory corticostriatal synapses are established through a selective and/or competitive process. It is important to determine whether this sparse innervation of SPNs by S1 is a characteristic shared with other projections. In particular, it will be interesting to test this hypothesis on the auditory projections targeting the posterior striatum, where neurons exhibit clear tone frequency selectivity (Guo et al., 2018).”

(3) In general, some of the figure legends are extremely brief, making many details difficult to infer. Similarly, some statistical analyses were either not carried out or not consistently reported.

We thank you for having taken the time to indicate where changes could benefit the paper. We have followed your recommendations.

**Reviewer #1 (Recommendations for the authors):**
A few limitations should be discussed in the manuscript:(1) The manuscript should mention that most corticostriatal synapses are formed at the dendritic spines of the SPNs, not their cell bodies. This is particularly important regarding the analysis and interpretation of the data in Figure 4.

Thank you for this comment. This characteristic is important with regards to a limitation of electrophysiological recordings. This is now discussed:

Line 275:

“The LSPS combined with glutamate uncaging mapped projections contained in the slice, intact from the presynaptic cell bodies to the SPN dendrites. Some cortical inputs targeting distal SPN dendrites may have gone undetected, either due to attenuation of synaptic events recorded at the soma or because distal dendritic branches were lost during slice preparation. Indeed, about 80 % of S1 synaptic contacts are distributed along dendrites (Sanabria et al., 2024). However, synapses located distally are proportionally rare (Sanabria et al., 2024), and our estimates suggest that the loss of S1 input was minimal (see Methods).“

Line 313:

[...],, we found that overlaps between the connectivity maps of SPNs were rare and, when present, involved only a small fraction of the connected sites. This indicates that neighboring SPNs predominantly integrated distinct inputs from the barrel cortex, although it is possible that overlapping inputs received in distal dendrites were not all detected”

(1) SPNs show up- and down-states in vivo, which were not mimicked by the present study since all cells were held at - 80 mV (Line 364) and recorded at room temperature (Line 368). It should be discussed how the conclusion of the present work may be affected by the up/down states of SPNs in vivo.

Thank you for raising this point. Indeed, our experimental conditions were not designed to capture the effects of network oscillatory activity. Instead, LSPS conditions were optimized to reveal monosynaptic connectivity between neurons in S1 and their postsynaptic targets. These optimizations include the use of a high concentration of extracellular divalents (4 mM Ca^2+^ and Mg^2+^) to generate robust yet moderate and spatially-restricted stimulations of cortical cells and reliable neurotransmitter release (Shepherd, Pologruto and Svoboda, Neuron 2003; 10.1016/s0896-6273(03)00152-1; in our study, see Fig. 1D and Suppl Fig. 2). Investigating the pre- and postsynaptic modulations of the corticostriatal coupling by up- and down-states would require specific conditions.

The conclusion now acknowledges that functional connectivity is subject to plasticity in general, line 358:

“The degree of input convergence onto SPNs could be modulated by plasticity, potentially enabling experience-driven reconfiguration of S1 corticostriatal coupling.”

(2) In addition to population-level integration (Line 337), sensory integration is likely to involve synaptic plasticity (like via NMDARs), which was not studied in the present work

Thank you for raising this point. Indeed, we agree that sensory integration is a complex process with a multitude of factors beyond connectivity patterns and synaptic strength. We also agree that both connectivity levels and synaptic strength can be modified by plasticity.

We modified our conclusion as follows, line 354:

“Since the inputs to a single SPN represent only a limited subset of whisker columns, a complete representation of whiskers could emerge at the population level, with each SPN’s representation complementing those of its neighbors (Fig. 7). These observations raise the hypothesis of a selective or competitive process underlying the formation of corticostriatal synapses. The degree of input convergence onto SPNs could be modulated by plasticity, potentially enabling experience-driven reconfiguration of S1 corticostriatal coupling. “

(3) The potential corticostriatal connectivity may be underestimated due to loss of axonal branches during slice resection, and this might contribute to the conclusion of "sparse connectivity". Whether the author has considered performing LSPS studies within the striatum (i.e., stimulating ChR2-expressing cortical axon terminals) and whether this experiment may consolidate the conclusion of the present work.

We appreciate the suggestion to employ Subcellular Channelrhodopsin-2-Assisted Circuit Mapping (sCRACM) to study the density of S1 spines on SPNs dendritic arbor. If ChR2 is broadly expressed in S1, this approach would likely increase spine detection, as spines contacted by presynaptic neurons located inside and outside the slice would now be activated. If ChR2 expression could be restricted to the whisker columns present in our preparation, enhanced detection could still occur, but in this case, it would reflect the activation of spines contacted by specific ChR2^+^ axonal branches that exit and re-enter the slice to form synapses on the recorded SPN. The anatomy of corticostriatal axonal arbors suggest convoluted axonal trajectories could be relatively rare (T. Zheng and C.J. Wilson, J Neurophysiol. 2001; 10.1152/jn.00519.2001; M. Lévesque et al., Brain Res. 1996; 10.1016/0006-8993(95)01333-4).

Moreover, it is important to remember that sCRACM does not generate connectivity maps between 2 structures, but maps of spines on dendritic arbors (Petreanu L.T. et al., Nature 2009; 10.1038/nature07709.). Precise localization of presynaptic cell bodies was key for the present study, as it enabled distinguishing between different connectivity patterns and between different degrees of convergence of inputs from adjacent S1 cortical columns present in the slice (schematized in Fig. 1). Distinguishing these inputs using the stimulation of axon terminals would require the possibility to express one distinct opsin in each whisker column (or each cortical layer, depending on the axis of investigation). This is an exciting perspective but the technology is not yet available to our knowledge.

To emphasize our reasons for using LSPS, we revised the final paragraph of the Introduction, line 69:

“LSPS enabled precise mapping of corticostriatal functional connectivity by identifying cortical sites where stimulation evoked synaptic currents in the recorded SPNs, thereby localizing the cell bodies of their presynaptic neurons. This approach allowed us to determine both the cortical column and layer of origin within the barrel field in the slice for each SPN input.”

**Reviewer #2 (Recommendations for the authors):**
(1) Figure 2F: SPN and cortical regions - both are shown in green. The distinction between the two would be clearer if SPNs were made a different color.

Done

(2) Figure 2H: Based on their data, the authors conclude that since EPSCs in SPNs had small amplitudes (~40pA), only one or a few presynaptic cortical neurons (< 5) were activated by uncaging. It is not clear how this number was estimated. Either this statement should be qualified with data or citations provided to support it.

We thank you for noticing it. We modified this part as follows, line 105:

“Based on known amplitudes of spontaneous and miniature EPSCs in SPNs (10-20 pA on average; Kreitzer and Malenka, 2007; Cepeda et al., 2008; Dehorter et al., 2011; Peixoto et al., 2016), this finding is consistent with the presence of only one or a few presynaptic cells (≤ 5) at each connected site of the map.”

(3) Figure 2I: The top graph is difficult to understand without already seeing the lower plot. Moving it below or to the side would help the reader follow the data more easily.

done

(4) Figure 3D: In Line 162, the authors state, " Furthermore, SPNs receiving input from a single column were often located near others receiving input from multiple ones (Figure 3D), reinforcing that the low functional connectivity with barrel columns in the slice was genuine in these cases." However, Figure 3D does not show spatial information about SPNs relative to each other. This data should be added or the statement adjusted to reflect what is shown in the panel.

Corrected as follows, line 167:

“Furthermore, SPNs receiving input from a single column were often located in slices where other cells received input from multiple ones (Fig. 3D), reinforcing that the low functional connectivity with barrel columns in the slice was genuine in these cases.”

(5) Figure 3F: Are the authors attempting to show how cluster number, cluster width, and connectivity gaps contribute to input field width? If so, this could be clarified by flipping the x- and y-axes so that the input field width is the y-axis in each case. Additionally, the difference between black and white points should be stated (or, if there is no difference, made to be the same). The significance of the dotted red line vs. the solid red lines should also be stated in the figure legend.

These plots illustrate how cluster number, cluster width, and ratio of connectivity gaps over total length vary as a function of input field width. As expected, wider input fields contain more clusters (top). However, the overall density of connected sites does not increase with input field width, as indicated by a higher ratio of connectivity gaps over total length (bottom).

This suggests the presence of a mechanism that regulates the connectivity level of individual SPNs (mentioned in the discussion). We prefer this orientation because the flipped one makes a cluttered panel due to different X axis labels. Symbols and lines were corrected. The correlation coefficients and statistics are now indicated in the panels and in the legend.

(6) Figure 3H: The schematic is very useful for highlighting the core conclusions and is greatly appreciated. The pie charts are a bit hard to see and could be replaced with the percentages stated simply as text within the figure. It would also help to label the panel as "Summary," so readers can quickly identify its purpose.

Done

(7) Figures 4B-D: To clarify the overall percentage, the maximum for the y-axis should be set to 100% in each panel.

Done

**Reviewer #3 (Recommendations for the authors):**
(1) Though mostly minor, several sentences/statements in the manuscript are confusing or overstated. For example:a. Lines 62-63: "Studies have found that inputs received by D1 SPNs were stronger than those received by D2 SPNs" is a broad statement that should be qualified.

We changed this sentence for:

“Electrophysiological studies have found that inputs received by D1 SPNs were stronger than those received by D2 SPNs, both in vivo and ex vivo (Reig and Silberberg, 2014 ; Filipović et al., 2019 ; Kress et al., 2013 ; Parker et al., 2016).”

b. Lines 118-119: "EPSCs evoked with stimulations in L2/3 to L5b had similar amplitudes (Figure 2H), suggesting that L5a dominated these other layers thanks to a greater connectivity with SPNs principally." Here, the word "connectivity" is vague and could easily be misunderstood. Connectivity could refer to the amplitude of corticostriatal EPSCs, which the authors stated are not different between L2/3-L5b. Presumably, connectivity here refers to % of connected SPNs, but for the sake of clarity, the authors should be more explicit, e.g,. "...L5a dominated the other layers because a larger fraction of SPNs received connections from L5a, rather than because L5a synapses were stronger."

We changed the sentence for (line 122):

“EPSCs evoked with stimulations in L2/3 to L5b had similar amplitudes (Fig. 2H), suggesting that L5a dominance over these other layers is primarily due to a higher likelihood of SPNs being connected to it, rather than to stronger synaptic inputs.”

c. In the Figure 4 legend, (A) says "Four example slices with 2 to 4 recordings. Same as in Figure 2A." Did the authors mean Figure 3A?

Done

d.Line 184: Should Figure 4B, C actually be Figure 4D?

Done

(2) Line 32: typo in Sippy et al. reference.

Done

(3) In Figure 2I, the label "dSPN" is confusing, as in the literature, dSPN often refers to the direct pathway SPN.

Done

(4) The y-axes in Figure 3C should be better labeled/explained.

Fig.3C. Median (red) and 25-75th percentiles (box) of cluster width and spacing, expressed in µm (left Y axis) and number of cortical columns (right Y axis). Labels have been changed in the figure.

(5) Lines 150-152: "...45 % of the input fields with several clusters produced no synaptic response upon stimulation." This wording is confusing. It can be inferred that the authors mean "no synaptic response in the gaps between clusters." However, their phrasing omits this crucial detail and reads as though those input fields produce no response at all.

We changed this sentence for (line 154):

“Strikingly, regions lacking evoked synaptic responses (i.e., connectivity gaps) made up an average of 45 % of the length of input fields with multiple clusters (maps collapsed along the vertical axis; Fig. 3F, bottom). “

(6) Lines 184-186: "DLS SPNs could receive inputs from the same domain in the barrel cortex and yet have patterns of cortical innervation without or little redundancy." This should be rephrased to "with little to no redundancy."

Done

(7) Lines 186-187: "They support a connectivity model in which synaptic connections on each SPNs..." should be revised to "connections to each SPN...".

Done